# Plastics and sustainable purchase decisions in a circular economy: The case of Dutch food industry

**Pedro Núñez-Cacho**[1]*, **Juan Carlos Leyva-Díaz**[2], **Jorge Sánchez-Molina**[3], **Rody Van der Gun**[4]

**1** Department of Business Organization, University of Jaén, Jaén, Spain, **2** Department of Chemical and Environmental Engineering, University of Oviedo, Oviedo, Spain, **3** Grupo de Investigación en Materiales Cerámicos - GITEC, University Francisco de Paula Santander, San José de Cúcuta, Colombia, **4** Department of Management, University of Granada, Granada, Spain

* pnunez@ujaen.es

**Data Availability Statement:** All relevant data are within the manuscript and its Supporting Information files.

## Abstract

Every day, society's concern over pollution caused by plastic waste grows greater. One of the most intensive sectors for the use of plastic is the food industry. Companies in this sector face the challenge of transitioning to a more sustainable and less intensive model of plastic use, respecting the principles established for a circular economy. Accordingly, one of the questions that industries tend to ask is whether sustainability will influence the consumer's purchase decision. To respond to this, the factors that determine a consumer's sustainable purchase decision in relation to the plastic and food industry have been analyzed in this paper. For this, a regression analysis was performed on a sample of Dutch consumers. The results show that the decision of purchase of the consumer of the Food Industry is conditioned by factors such as age, sustainable behavior, knowledge of the Circular economy and the perception of usefulness of plastic.

## Introduction

Day by day, the debate on the sustainability of the planet is becoming more relevant. Concerns about the effects of $CO_2$ emissions and uncontrolled generation of waste are causing companies to consider a new scenario in which the consumer is increasingly selective in making a purchase decision, conditioned by aspects related to sustainability [1, 2].

As a result, there is widely increasing attention on the problems caused by the generation of plastic waste, with society trying to replace unsustainable habits. Plastic waste also severely causes marine pollution and has become a large-scale problem after only half a century of widespread plastic use [3]. The Ellen MacArthur Foundation (EMF) stated that most of the plastic packaging produced ends up in landfills or oceans: "With 8 million tons of plastic that reach the ocean every year, we urgently need to rethink the way we manufacture, use and reuse plastics." Plastic waste is a significant threat to the environment due to its resistance to photooxidative, thermal, mechanical and biological processes [4–6]. In the 1950s, the global

**Funding:** The author(s) received no specific funding for this work.

**Competing interests:** The authors have declared that no competing interests exist.

**Abbreviations:** AVE, Average of variance extracted; CE, Circular Economy; EMF, Ellen MacArthur Foundation; HDPE and LDPE, respectively high and low density polyethylene; KMO, Kaiser-Meyer-Olkin Index; PET, Polyethylene terephthalate; PP, Polypropylene; PS, Polystyrene; PVC, Polyvinyl chloride; TPB, Theory of Planned Behavior.

production of plastic was only 2 million tons per year. By 2018, the world had produced 359 millions of metric tons of plastic. The problem is growing, and the use of plastic will double within the next 20 years [7].

One of the main consumers of plastic is the food industry, a sector where plastic is the most common material used for packaging. Businesses in this sector worry about packaging because changes or new requirements may result in an increase in the cost of production and investment in new equipment, and the economic performance of food manufacturers can be seriously altered in terms of productivity [8]. In 2018, Greenpeace made a list of the ten largest waste generators based on a cleaning project for the collection and classification of plastics according to the brands that used them. Nine of the ten most polluting brands were large multinationals operating in the global food industry.

Undoubtedly, plastic food packaging has numerous advantages, such as its low weight, which translates into lower logistics costs, and the durability and cost of the material, which is lower than that of other materials [9]. Thus, alternative packaging materials require twice as much energy for the production process as plastic. Durability is one of the greatest assets of this material. According to the US National Park Service, a plastic bottle can take 450 years to decompose in nature.

The challenge we face is in changing the way we handle our plastics. We currently manufacture, use and dispose of plastics, following what is known as a linear economy. Today, only 14 percent of global plastic containers are collected for recycling [10]. Consequently, most of the plastic packaging is incinerated or terminated in landfills or oceans. However, plastic food containers have the potential to be recycled in a closed circuit. This is why it is worth studying the transition to a circular economy (CE). 'The CE is a general term covering all activities that reduce, reuse, and recycle materials in production, distribution, and consumption processes [11]. CE represents a fundamental alternative to the linear take-make-consume-dispose economic model [12]. The transition to the new mode find inertias [13], however the process could be sped up by different drivers issued by the company's main stakeholders. Implementing a CE over the long term will depend on consumers' perception of added value [14]. If we analyze the behavior of today's consumers, it is clear that they are more informed and take into account the aspects of environmental sustainability when making a purchase decision.

A knowledge gap lies in how customers make purchase decisions based on the sustainability of the desired products, making this research relevant. This is especially so because the CE model is applied to the food industry in the Netherlands, which is a sector responsible for a large share of the global use of plastic. Plastic is the most common material for food contact [15]. Therefore, environmentally sustainable improvements are required within this industry. Knowing the factors that affect a consumer's purchase decision can drive the food industry's transition to a CE model. Based on the problem definition and aim of this study, the following research question was formulated: What characteristics of the consumer influence a sustainable purchase decision? To answer this main research question, some sub-questions were posed and had to be answered:

**RQ$_1$**. Is a sustainable purchase decision influenced by a consumer's attitude or behavior toward sustainability?

**RQ$_2$**. Does a consumer's knowledge of the CE influence their purchase decision?

**RQ$_3$**. Do demographic aspects such as age influence the sustainable purchase decision?

## Theoretical framework and hypotheses

### Consumer purchase decision

The purchase decision is the behavior shown by the consumer during the process of buying goods and services. This process includes the analysis, evaluation, acquisition and use of a good or service [16, 17]. When a consumer uses decision criteria related to sustainability for the analysis, evaluation and selection of a product, it is referred to as a sustainable purchase decision.

The Theory of Planned Behavior (TPB) points out that consumers are not rational in their behavior when they consider buying a product. The decision-making process is influenced by beliefs, habits, knowledge and social norms [18, 19]. Ajzen [20] stated that behaviors are derived from intentions. The intention behind a behavior has three determinants: attitude of behavioral belief, subjective norms of normative belief and behavioral control.

Those who feel guilty about their behavior will sacrifice their own interest for the welfare of others [21]. This author distinguishes three antecedents to predict consumer behavior. First, the activation of a behavior begins with awareness of the consequences and attribution of responsibility. According to Park et al. [22], it is difficult for one to feel obliged to perform a certain behavior when unaware of its consequences. Only knowledgeable consumers can take responsibility. If consumers know that recycled packaging can solve environmental problems, they will be motivated to take responsibility for recycling. Further, knowledge of rules that favor pro-environmental altruistic behaviors, of which recycling is a type [23, 24], and whether people have had personal past behaviors associated with recycling are important factors that determine consumers' purchasing behaviors. Tonglet et al. [25] pointed out that a person's attitude toward the performance of a particular behavior affects their intention to carry out said behavior, which ultimately influences their actual behavior [26–29].

Thus, the position and behavior of the food industry with respect to sustainability and recycling are influenced by the beliefs of others and social pressure. In addition, a consumer's intention to recycle is expected to increase when other influential people, such as family and friends, behave in a way that is beneficial to the environment (i.e., recycle materials) and believe that recycling is not a hard task [22].

On the other hand, what is known as "perceived behavior control" refers to the extent to which a person considers it easy or difficult to exhibit a behavior within a certain context. Studies on recycling behaviors by Boldero [30], Davies et al. [31] and Mannetti et al. [32] show that the three determinants of intention that we have highlighted are positively related to the intention to recycle.

Environmental worldviews are collective beliefs and values that give people an idea of how the world works. Environmental worldviews can deeply inform a person's understanding and perception [33]. Behavior and perceptions of consumers are essential for the success of a CE [34]. Environmental worldviews based on consumer attitudes are among the most important variables that influence environmental action [35–37]. Worldviews influence decisions and involve people in thinking of solutions to address environmental problems, such as the use of plastic food containers. However, people may fail in their behaviors to improve their own interests or those of society because of a lack of relevant information [38].

Pro-environmental behaviors of consumers are influenced by environmental worldviews and personal effectiveness. According to Chen et al. [39], personal effectiveness is the individual judgment of a person's ability to perform a particular behavior. Personal efficacy beliefs affect a person's individual choices, tasks, efforts and persistence level. Feelings of personal efficacy influence the motivation of an individual to participate in pro-environmental behaviors. People evaluate their behaviors based on their contributions to the resolution of

environmental problems. An individual with low personal effectiveness is not likely to act with great effort or persist in the face of environmental problems. Thus, there is a correlation between personal effectiveness and concern for the environment [40]. In addition, environmental worldviews influence personal effectiveness. A person who does not consider plastic food packaging to be an environmental problem will not be involved in the same. Individuals feel responsible and experience personal effectiveness in solving environmental problems that they are well informed of [41]. Here, environmental knowledge, which changes the vision of the environmental world, is essential to induce a particular pro-environmental behavior. Kellstedt et al. [42] provided a high level of information on the ecological impact of human behavior toward a greater degree of personal effectiveness. Therefore, we expect that a consumer concerned with the impact of plastic on the environment would have a willingness to formulate in a sustainable way. Hence, we formulated the following hypothesis.

$H_1$: *Environmental concerns about plastic use positively influence a consumer's sustainable purchase decision.*

Stern [43] provided a theoretical understanding of the role of values in pro-environmental behavioral intentions and real behaviors. This is related to a consumer's belief about whether plastic packaging damages the environment. If consumers think that not recycling plastic containers causes problems, they have an incentive to recycle. Respecting the rules that drive purchasing decisions, a consumer's personal rules refer to the degree to which the consumer feels a personal obligation to adopt a certain pro-environmental behavior. Further, the Reasonable Person Model (RPM) developed by Kaplan and Kaplan [44] highlighted the role of reasonableness in civil behavior. The RPM is a useful framework to understand humans, their actions and their convictions as well as investigate how the environment can help make the best of people. A person's behavior toward the environment can be irrational, such as using resources (not renewable) to make a profit and leaving nothing for future generations. Access to information plays an important role in determining one's environmental behaviors. "People are more reasonable, cooperative, helpful and satisfied when the environment supports their basic information needs" [44]. Resources are scarce and we need a proper management [45].

There are three domains of information needs. First is the construction of models, which are simplified versions of reality that are stored in the minds of people [46]. Information facilitates one's understanding of the world and reduces confusion. Humans are animals based on information—to survive we need to explore information and learn about the environment. The second domain involves effectiveness, which deals with the ability of people to manage the abundant information available. This also includes handling the available information correctly. The ability to do so requires the right skills and motivation. Humans need to have a clear mind and the right skills to feel effective. Information is the basis of human functioning; one needs to manage the constant flow of information to feel effective. Feeling competent depends on knowing how to do things and knowing what is possible. The last source of information, meaningful action, refers to the need to be an active person and receive all the accessible information. Humans are social actors and, therefore, desire their actions and personal behaviors are important [46]. We can, therefore, expect that when a consumer decides to buy a food product, they value aspects related to sustainability and recycling, thereby conditioning their purchase decision. Therefore, we propose the following hypothesis:

$H_2$: *Consumer behaviors related to sustainability positively influence sustainable purchase decisions in the food industry.*

A consumer's decision in the food industry is conditioned by personal effectiveness. According to Chen et al. [39], personal effectiveness is an individual's judgment of their own

ability to perform a particular behavior. A person's beliefs about their personal effectiveness affect their individual choices related to tasks, effort and persistence. Feelings of personal efficacy influence the motivation of an individual to participate in pro-environmental behaviors [47]. People evaluate their behaviors based on the importance of their contributions to the resolution of environmental problems. An individual with low personal effectiveness is not likely to act with great effort or persist in the face of environmental problems. According to Kellstedt et al. [42], a high level of information on the ecological impact of human behavior leads to a greater degree of personal effectiveness and effort in making decisions related to sustainability. Here, environmental knowledge, including CE, is essential for a particular pro-environmental behavior and the decision to purchase sustainable products.

Therefore, we propose the following hypotheses:

$H_3$: *The knowledge of CE principles positively influences sustainable purchase decisions.*

$H_4$: *The knowledge of sustainability principles positively influences sustainable purchase decisions.*

## Plastic in the food industry

According to the Federal Indian Chamber of Commerce and Industry, plastics are currently the most used packaging materials (42%), followed by paper board (31%), metals (15%), glass (7%) and other materials (5%). This is probably due to the high cost-to-performance ratio, light weight, and convenience attributed to plastics.

Further, the World Packaging Organization (WPO) [48] estimated that the food industry accounted for 54% of all the plastic packaging during the period 2003–2009. There are many examples of plastic food packaging, i.e., bottles, trays, pots, foils, bags, cups, pouches, bowls and others [49]. Among the different plastics in food packaging, it should be highlighted that polyethylene terephthalate (PET), polypropylene (PP), high and low density polyethylene (HDPE and LDPE, respectively), polystyrene (PS) and polyvinyl chloride (PVC) are plastic multilayers that combine several kinds of plastic layers [49, 50]. In this regard, it should be noted that plastic packaging is primarily manufactured from non-renewable raw materials and are currently a small market share of bioplastics [51].

According to Plastics Europe [52], around 30 kg of plastic packaging waste was generated per inhabitant per year in Europe in 2015. Plastics are usually discarded in landfills or incinerated for energy generation, which causes problems with space availability and air pollution. Only a low amount of plastic is recovered by recycling [53]. Recycling can be performed through mechanical and chemical processes [54]. Mechanical recycling processes are the most common because of economic and environmental issues, including the operations of cleaning, grinding, re-melting and re-granulating [49, 55]. Due to this, the Plastics Recycling Regulation regulates the use of recycled plastics in food contact materials.

The current packaging designs have begun to include recyclable and recycled plastics, but recycling is conditioned by the safety evaluation of recycling processes and economic assessment of costs for collection, separation, cleaning, reprocessing and transportation [49, 50].

Consumers disagree about the severity of environmental problems generated by plastic food containers. Different opinions derive from different environmental worldviews. Environmental worldviews are collective beliefs and values that give people an idea of how the world works. They can profoundly influence people's understanding and perception of what their role in the environment should be and the right and wrong environmental behaviors [33]. Environmental worldviews based on consumer attitudes are among the most important

variables that influence environmental action [35–37]. However, according to Maibach et al. [38], people fail in their behaviors to improve their own interests or those of society because they do not have relevant information. Thus, knowledge of the various aspects related to sustainability influences a consumer's behavior and sustainable purchase decisions. There is a correlation between personal efficacy and concern for the environment [40]. In addition, environmental worldviews influence personal effectiveness. For instance, a person who does not consider the environmental problems due to plastic food packaging will not be involved in the recycling process. People feel responsible for and experience personal effectiveness in solving environmental problems when they are knowledgeable about them and have enough information [41]. So, we formulate the next hypotheses:

$H_5$: *A person's perception of the utility of plastic positively influences their sustainable purchase decisions.*

$H_6$: *A person's perception of the harmful nature of plastic negatively influences sustainable purchase decisions.*

## Personal factors and sustainable purchase decisions

As previously discussed, subjective norms pressure individuals to modify their behaviors in aspects such as purchase decisions. These influences mark acceptable and unacceptable behaviors. Therefore, influencing factors encourage or discourage buying behavior [29]. Each individual, depending on their environment, would receive be influenced by something. According to [56], the buying behavior of a consumer is influenced by cultural, social, personal and psychological factors. Therefore, consumer behavior is understood as a part of human behavior. By studying it, researchers can predict how consumers will behave in the future when making purchasing decisions. In addition, personal characteristics influence a person's decisions, habits, interests and behaviors [17]. Some influential aspects include age, gender, culture and other personal characteristics. An older person may be perceived to exhibit different consumption habits than a younger person. Consumers have different needs based on their age group, beliefs and attitudes. Therefore, depending on the environment, age, gender and other personal characteristics, the consumer can be expected to have differentiated behaviors while making sustainable purchase decisions, resulting in the following hypothesis.

$H_7$: The age of the consumer positively influences the purchase decision.

## Methods

### Sample

To achieve the objective of this research study on the behaviors of food consumers toward the plastic packaging of food and CE, a web-based questionnaire was administered and information on the knowledge, dispositions and behaviors of consumers in the food industry was obtained. It was supplied to a representative sample of food consumers in the Netherlands. The questionnaire was designed using the Likert multiple-choice question scale, which appears in annex 1, and it was administered through email and social media networks. 220 valid responses were received. For this research, a questionnaire has been created to investigate consumer behavior. The questionnaire includes a combination of Likert scale, multiple choice

**Table 1. Age of respondents.**

| Age | Frequency | Percent |
|---|---|---|
| 18–24 | 65 | 29.5 |
| 25–34 | 80 | 36.4 |
| 35–44 | 13 | 5.9 |
| 45–54 | 33 | 15 |
| 55–64 | 28 | 12.7 |
| 65–74 | 1 | 0.5 |

questions and open ended questions. The objective is to clarify the behavior of the food consumer towards plastic food packaging and the CE. The sample is made up of European food-consuming citizens. The sample size is (N = 220). All questions are presented in the appendix. The questionnaire is distributed to European food consumers, through WhatsApp, Facebook, LinkedIn and is sent by email directly to respondents. These are accessed, accessed and completed an online questionnaire between the months of May and June 2019. The response rate was 46%. The age range of the sample was from 17 to 66 years. Table 1 presents information on the sample composition.

Information on the other demographic aspects of the sample, such as income levels, are shown in Tables 2 and 3 presents information about the interviewees' knowledge of aspects related to the circular economy, and Table 4 shows the concerns related to the environmental situation.

## Methods

Several procedures were used to quantify the information. First, an exploratory factor analysis was performed to determine the factors under which the variables could be grouped. The factors analyzed were environmental concerns, plastic harms, plastic utility, sustainable behavior, sustainable purchase decision, sustainability knowledge and CE knowledge. Once the factor analysis was carried out, the average variance extracted (AVE) was checked and found to be sufficient in all the cases. Subsequently, a Cronbach alpha analysis was performed to determine

**Table 2. Income levels of the sample.**

| Income levels | Frequency | Percent |
|---|---|---|
| Between €0 and €10.000 | 72 | 32.7 |
| Between €10.000 and €20.000 | 34 | 15.5 |
| Between €20.000 and €30.000 | 22 | 10 |
| Between €30.000 and €40.000 | 37 | 16.8 |
| €40.000 or higher | 55 | 25 |

**Table 3. Degree of knowledge of CE.**

| Knowledge about CE | | |
|---|---|---|
| Degree | Frequency | Percent |
| Low | 40 | 18.2 |
| Medium | 79 | 35.9 |
| High | 101 | 45.9 |
| Total | 220 | 100 |

**Table 4. Concerns about the impact of plastic on the environment.**

| Concern | Frequency | Percent |
|---|---|---|
| 1 | 1 | .5 |
| 2 | 5 | 2.3 |
| 3 | 17 | 7.7 |
| 4 | 12 | 5.5 |
| 5 | 32 | 14.5 |
| 6 | 55 | 25.0 |
| 7 | 98 | 44.5 |
| Total | 220 | 100.0 |

(1 Very low / 7 very high)

the reliability of the scale, which also verified this aspect. The results of these analyses are given in Table 5.

Following this, to check the relationships presented in the research model, an inferential as well as descriptive regression analysis was performed using the SPSS software. The following variables were used: The dependent variable was the sustainable purchase decision. The independent variables were the individual's sustainable behavior, sustainable attitude, level of knowledge about sustainability and CE, concern for environmental problems, perception of the negative impact of plastic food packaging on the environment, income level, age and living situation.

## Measuring scales

**Dependent variable: Sustainable purchase decision.** A purchase decision involves the behavior shown by the decision-making units in the purchase, use and disposal of goods or services [16]. When a consumer introduces sustainability into the decision-making process, it is called a sustainable purchase decision. Four variables related to the importance that a consumer places on aspects related to sustainability were included, such as recycling, repair and degradability, among others. The variables were measured using a Likert-type scale from 1 to 7.

**Independent variables.** According to the proposed research model, the following were the independent variables.

**Table 5. Exploratory factor analysis and reliability of the scales.**

| FACTOR | KMO | Bartlett's Test of Sphericity | | | | | Reliability Statistics |
|---|---|---|---|---|---|---|---|
| | | A. Chi-Square | df | Sig. | AVE | | Cronbach's Alpha |
| Environmental concerns | 0.934 | 1517.164 | 28 | 0 | 73.70% | | 0.948 |
| Plastic harmful | 0.611 | 98.978 | 1 | 0 | 80.32% | | 0.753 |
| Plastic utility | 0.596 | 66.841 | 1 | 0 | 75.72% | | 0.679 |
| Sustainable behavior | 0.674 | 130.520 | 3 | 0 | 67.50% | | 0.613 |
| Sustainable purchase decision | 0.733 | 234.727 | 6 | 0 | 61.03% | | 0.762 |
| Sustainability knowledge | 0.719 | 147.940 | 6 | 0 | 53.77% | | 0.695 |
| Circular economy knowledge | 0.633 | 60.619 | 3 | 0 | 54.69% | | 0.685 |

*(1) Level of knowledge about CE.* For its measurement, a factor composed of three variables was employed based on specific aspects and basic principles of CE, such as knowledge about what CE, cradle-to-cradle principles and closed cycle recycling are.

*(2) Level of knowledge about sustainability.* This factor consisted of four variables related to sustainability, i.e., knowledge about the waste hierarchy pyramid, extended producer responsibilities, social responsibilities of companies and life cycle analysis.

*(3) Environmental concerns.* The factor involved the grouping of eight variables related to the concern of consumers, including aspects such as the amount of plastic waste produced, the total waste sent to landfills, water pollution and concerns about biodiversity, resources, pollution, global warming, etc.

*(4) Sustainable behavior.* This factor involved three variables related to the actions carried out by the interviewees to improve sustainability. It included being aware of plastic pollution, reduction in the intensity of the use of plastic bags for food transport and reduction in the amount of plastic used in packaging.

*(5) Plastic.* This research study also included two specific factors related to plastic. The first was the perception of its utility, and the second was the perception of it as harmful material. These factors were measured using four variables related to this topic, such as its advantages, its usefulness, the perception of the damage it generates and the environmental inconveniences that it causes.

Two control variables were introduced: life situation of the respondents and genre. The variables were measured using a Likert scale of 1 to 7.

**Data analysis.** We used a factorial exploratory analysis and the Cronbach alpha to assess the validity of the measurement scales and factors used. The statistical analysis of the factors is shown in Table 5, including the calculation of the Kaiser-Meyer-Olkin measure of sampling adequacy (KMO), the Bartlett test and AVE. The results show that the factors were properly designed, and the scales of measurement were reliable.

Table 6 presents information on the correlation of the factors. The correlation between sustainable purchase and CE knowledge, sustainability behavior and age was strongly positive and significant, providing preliminary evidence for the hypotheses formulated. In the cases of high correlation, we checked the multicollinearity of the factors to avoid dependent cases.

The methodology used for the analysis of the relationships raised in the research model involved a hierarchical multiple regression analysis, which is also useful for predicting the

**Table 6. Correlation analysis.**

| | 1 | 2 | 3 | 4 | 5 | 6 | 7 | 8 | 9 | 10 | 11 |
|---|---|---|---|---|---|---|---|---|---|---|---|
| **Sustainable purchase** | 1 | | | | | | | | | | |
| **Living** | 0.059 | 1 | | | | | | | | | |
| **Educational level** | 0.137 | 0.021 | 1 | | | | | | | | |
| **Sustainability beh.** | 0.208 | 0.243 | 0.020 | 1 | | | | | | | |
| **CE knowledge** | 0.170 | -0.197 | 0.053 | 0.144 | 1 | | | | | | |
| **Sustainability knowl.** | 0.156 | 0.049 | 0.159 | 0.183 | 0.476 | 1 | | | | | |
| **Environment concerns** | 0.014 | 0.187 | 0.013 | 0.379 | 0.126 | 0.056 | 1 | | | | |
| **Plastic utility** | -0.168 | 0.067 | -0.059 | -0.103 | -0.005 | 0.081 | -0.173 | 1 | | | |
| **Age** | 0.400 | 0.066 | 0.289 | 0.103 | 0.030 | 0.088 | -0.014 | -0.068 | 1 | | |
| **Genre** | -0.164 | -0.014 | 0.024 | -0.007 | 0.230 | 0.147 | -0.048 | 0.295 | -0.265 | 1 | |
| **Plastic harmful** | 0.070 | 0.055 | 0.14 | -0.291 | 0.056 | 0.070 | -0.217 | 0.355 | 0.122 | 0.127 | 1 |

**Table 7. Models and variables.**

| Variables | Model 1 | Model 2 | Model 3 | Model 4 |
|---|---|---|---|---|
| Age | .283 | .271 | .270 | .264** |
| Sustainability Behavior | | .181 | .160 | .146** |
| CE Knowledge | | | .135 | .137* |
| Plastic utility | | | | .127* |
| Constant | .699 | 1.6990 | 1.5670 | 1.477 |
| N observations | 212 | 212 | 212 | 212 |
| Sig. | .000 | .000 | .000 | .000 |
| F | 40.06¨ | 24.25¨ | 18.02¨ | 14.83¨ |
| R | .400 | .434 | .455 | .472 |
| $R^2$ | .160 | .188 | .207 | .223 |
| $R^2$ change | .160 | .028 | .018 | .016 |
| Sig. F change | .000 | .008 | .029 | .040 |

\* significant p < .05

\*\* significant p < .01

Excluded variables: Living, educational level, environmental concerns,
sustainability knowledge, plastic harmful

dependent variable, and gradual regression. Prior to this, the data was analyzed to assess the suitability of the analysis. First, the data requirements were analyzed to perform the multiple regression analysis, such as the continuity of the dependent and independent variables. We also analyzed the independence of the observations, which was tested by means of the Durbin–Watson statistical test, the linearity, which was deduced through a scatter plot of the variables and the homoscedasticity, non-multicollinearity and, finally, normal distribution of the residues, verified through the Durbin–Watson statistical test, obtaining a score of 1.692.

Once these requirements were verified, we performed the regression analysis in successive steps that generated four iterations. Table 7 shows the variables that were introduced in models 1, 2, 3 and 4. According to the Anova test (Table 8), the four models used were statistically significant (p > .0005). The model fit appears in the Tables 9 and 10. The predictive variables were age, attitude towards sustainability, knowledge of the CE and perception of the usefulness of plastic. In the successive models, the variables given in Table 7 were excluded.

The results obtained from the regression analysis allowed us to confirm hypothesis two, which relates the sustainable purchase decision with sustainable behavior, and hypothesis number three, which relates it to the knowledge of CE. The results also confirm hypothesis number five, which relates sustainable purchase decisions with the perception of plastic utility. Finally, hypothesis seven, which relates age with sustainable purchasing behaviors, was also confirmed. Thus, a model that allows us to predict the relevance of these variables in consumer behavior was configured. However, hypotheses one, four and six were not confirmed and were excluded from the model.

The predictive equation of the sustainable purchase decision is:

$$SPD = 1.477 + .264 \ A + .146 \ S + .137 \ C + .127 \ P$$

Where:
A = Age
S = Sustainability behavior
C = CE knowledge

**Table 8. ANOVA[e].**

| Model | | Sum of Squares | df | Mean Square | F | Sig |
|---|---|---|---|---|---|---|
| 1 | Regression | 33.139 | 1 | 33.139 | 40.069 | .000[a] |
| | Residual | 173.680 | 210 | 0.827 | | |
| | Total | 206.819 | 211 | | | |
| 2 | Regression | 38.960 | 2 | 19.48 | 24.254 | .000[b] |
| | Residual | 167.859 | 209 | 0.803 | | |
| | Total | 206.819 | 211 | | | |
| 3 | Regression | 42.763 | 3 | 14.254 | 18.072 | .000[c] |
| | Residual | 164.056 | 208 | 0.789 | | |
| | Total | 206.819 | 211 | | | |
| 4 | Regression | 46.088 | 4 | 11.522 | 14.839 | .000[d] |
| | Residual | 160.730 | 207 | 0.776 | | |
| | Total | 206.819 | 211 | | | |

[a]. Predictors: (Constant), age

[b]. Predictors: (Constant), age, sustainability behavior

[c]. Predictors: (Constant), age, sustainability behavior, CE knowledge

[d]. Predictors: (Constant), age, sustainability behavior, CE knowledge, plastic utility

[e]. Dependent variable: Sustainable purchase decision

P = Plastic utility

SPD = Sustainable purchase decision

## Discussion

In recent years, companies have evolved toward a sustainable governance model, aware that consumers' concerns about environmental issues are growing. The increase in the use of

**Table 9. Model summary.**

| Model | R | R Square | Adjusted R Square | Std. Error of the Estimate |
|---|---|---|---|---|
| 1 | .400[a] | 0.160 | 0.156 | 0.90942046 |
| 2 | .434[b] | 0.188 | 0.181 | 0.89618740 |
| 3 | .455[c] | 0.207 | 0.195 | 0.88810497 |
| 4 | .472[d] | 0.223 | 0.208 | 0.88117842 |

**Table 10. Model summary: Change statistics.**

| Model Summary[e] | | | | | | |
|---|---|---|---|---|---|---|
| Model | Change Statistics | | | | | Durbin-Watson |
| | R Square Change | F Change | df1 | df2 | Sig. F Change | |
| 1 | 0,16 | 40,069 | 1 | 210 | 0 | |
| 2 | 0,028 | 7,247 | 1 | 209 | 0,008 | |
| 3 | 0,018 | 4,821 | 1 | 208 | 0,029 | |
| 4 | 0,016 | 4,283 | 1 | 207 | 0,04 | 2,012 |

[a]. Predictors: (Constant), age

[b]. Predictors: (Constant), age, sustainability behavior

[c]. Predictors: (Constant), age, sustainability behavior, CE knowledge

[d]. Predictors: (Constant), age, sustainability behavior, CE knowledge, plastic utility

plastic in recent decades is one of the factors that conditions consumer behavior. In fact, in the sample analyzed for this study, 69.5% of the consumers were worried about the impact of plastic on the environment.

The food industry is included in the list of major plastic generators, according to Greenpeace data. TPB states that the purchase decision process is conditioned by the consumer's beliefs, habits, knowledge and norms. Therefore, we hope that a consumer concerned about sustainability will incorporate their knowledge and beliefs of CE aspects in the purchase decision process. In fact, a consumer is typically willing to sacrifice their own interests for the welfare of others [21]. In this context, companies in the food industry must be aware of the impact of sustainability on consumer purchase decisions.

Through our study, we were able to verify, in response to our first research question, that the sustainable purchase decision is conditioned by different consumer characteristics. Those consumers who follow sustainable behaviors in their ordinary lives, habits and customs will make sustainable purchase decisions in the food industry as well. This was confirmed through our research for the second hypothesis.

In the TPB arguments, we also made a reference to the fact that the decision-making process is conditioned by consumer knowledge. This is in accordance with to our second research question about whether the knowledge conditioned the purchase decision. Thus, the 82% of the sample participants knew the CE. Further, the results from the regression analysis allowed us to test the hypothesis that those consumers who have knowledge of the CE would opt for a sustainable purchase decision. Thus, we verified the theoretical arguments for the TPB and added a new explanatory factor for the sustainable purchase behavior of consumers related to the use of plastic packaging in the food industry.

The third research question in our work was whether the demographic aspects of the consumer, such as income level or age, condition their sustainable purchase decisions. Our research included a profile of the interviewees, and the predominant age range of 18–34 years accounted for 66% of the total. The results of the regression analysis allowed us to confirm the seventh hypothesis, establishing the influence of age on sustainable purchase decisions, i.e., younger buyers make more sustainable purchasing decisions in the food industry.

We had also raised aspects related to plastic as the determinants of sustainable purchase decisions in the food industry. With this hypothesis, the perception of the utility of plastic was incorporated into the consumer's purchase decision. There is packaging used by the food industry that a consumer would perceive as useful and necessary for the conservation and transport of food and others that are not. According to the results of the investigation, the perception of usefulness of the packaging also conditions the decision of sustainable purchase. The consumer will favor those products wherein the plastic packaging is seen as useful or essential compared to those which aren't perceived as such. With this factor, we introduced the perception of utility as the fourth condition for sustainable purchase decisions, thereby forming the final model (Fig 1).

## Practical implications

With regard to the practical implications for managers of food industry companies, we propose several conclusions that may be useful for practitioners. The use of plastic packaging models perceived by the consumer as unhelpful will make it difficult to sell the product and unsustainable in the long term. Younger consumers are more demanding when it comes to sustainability, so the products focused on this niche market should be the most sustainable in terms of their plastic packaging. A trained, knowledgeable consumer of the CE will also have higher expectations of sustainability in the packaging of food products. Additionally, if the

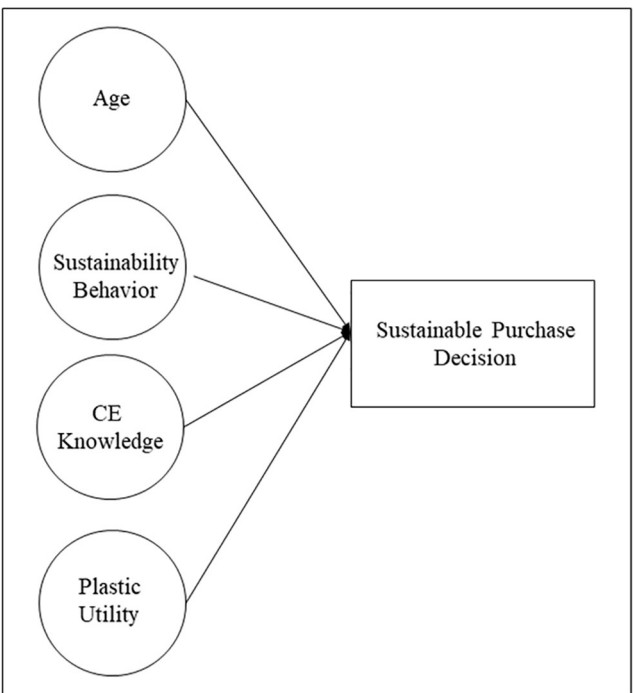

**Fig 1. Factors influencing sustainable purchase decisions.**

consumer follows sustainable behaviors in other aspects of life, they will be more aware of environmental damages and look for products with packaging that minimizes the ecological impact.

## Limitations and future perspectives

This study has certain limitations. The use of questionnaires to gather information gives rise to specific limitations due to the subjectivity involved in the use of this tool. When questionnaires are used, the researcher does not directly approach the phenomenon under study, and the respondents have a margin of freedom of interpretation that can distort the target set. In addition, the respondents' responses may reflect their own prejudices, since many items are based on their perceptions. To avoid this problem, we turned to secondary sources of data on organizational performance. However, the characteristics of the sample may also condition the study, so it is desirable to broaden it further.

Another limitation originates from the horizontal nature of the investigation, as this survey on plastic in the food industry was conducted during a single period of time. It would be worth analyzing sustainable purchasing behavior from an evolutionary perspective.

The limitations and depth of the study lead to a series of future research proposals as given below. We believe that it would be worth elaborating on an explanatory model of a consumer's sustainable purchase decision for a longer period of time instead of a specific moment and using a longitudinal analysis to observe the evolution of the variables. We can also consider the possibility of integrating variables that moderate the impact of the explanatory variables into the model or expanding the list of variables that may condition sustainable purchasing behaviors.

## Conclusion

In summary, the food industry faces a new scenario marked by sustainability and must make substantial changes to packaging to achieve greater competitiveness in the market. Our study indicates that there are several market niches that the food industry must serve, meeting their expectations for sustainable packaging. These will be the first steps in the transition of this industry towards a more sustainable packaging model.

## Supporting information

**S1 Questionnaire.**
(PDF)

## Author Contributions

**Conceptualization:** Pedro Núñez-Cacho, Jorge Sánchez-Molina, Rody Van der Gun.

**Data curation:** Rody Van der Gun.

**Formal analysis:** Pedro Núñez-Cacho, Juan Carlos Leyva-Díaz, Jorge Sánchez-Molina, Rody Van der Gun.

**Investigation:** Pedro Núñez-Cacho, Juan Carlos Leyva-Díaz, Rody Van der Gun.

**Methodology:** Pedro Núñez-Cacho, Juan Carlos Leyva-Díaz.

**Project administration:** Juan Carlos Leyva-Díaz.

**Resources:** Juan Carlos Leyva-Díaz, Jorge Sánchez-Molina.

**Software:** Pedro Núñez-Cacho, Jorge Sánchez-Molina.

**Supervision:** Pedro Núñez-Cacho, Jorge Sánchez-Molina.

**Validation:** Pedro Núñez-Cacho, Jorge Sánchez-Molina.

**Visualization:** Jorge Sánchez-Molina.

**Writing – original draft:** Pedro Núñez-Cacho, Jorge Sánchez-Molina.

**Writing – review & editing:** Pedro Núñez-Cacho, Jorge Sánchez-Molina.

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
