## [Decision Letter · Decision Letter 0]

9 Jun 2020

PONE-D-20-10526

Plastics and Sustainable Purchase Decisions in a Circular Economy: The Case of the Holland Food Industry

PLOS ONE

Dear Dr. Pedro,

Thank you for submitting your manuscript to PLOS ONE. After careful consideration, we feel that it has merit but does not fully meet PLOS ONE’s publication criteria as it currently stands. Therefore, we invite you to submit a revised version of the manuscript that addresses the points raised during the review process.

We look forward to receiving your revised manuscript.

Kind regards,

Prakash Kumar Sarangi, PhD

Academic Editor

PLOS ONE

Journal Requirements:

3. In your Methods section, please provide additional information about the participant recruitment method and the demographic details of your participants. Please ensure you have provided sufficient details to replicate the analyses such as: a) the recruitment date range (month and year),  b) a description of how participants were recruited, and c) descriptions of the social media platforms where participants were recruited and other methods for recruiting participants. Please also discuss whether the questionnaire was anonymized when you received the data.

'No'

'No'

6. Please include a copy of Table 7 which you refer to in your text on page 13.

7. We note you have included tables to which you do not refer in the text of your manuscript. Please ensure that you refer to Tables 9. 10 and 11 in your text; if accepted, production will need this reference to link the reader to the Tables.

Additional Editor Comments (if provided):

Reviewers' comments:

Reviewer's Responses to Questions

**Comments to the Author**

1. Is the manuscript technically sound, and do the data support the conclusions?

Reviewer #1: Yes

Reviewer #2: Yes

2. Has the statistical analysis been performed appropriately and rigorously? 

Reviewer #1: Yes

Reviewer #2: Yes

3. Have the authors made all data underlying the findings in their manuscript fully available?

Reviewer #1: Yes

Reviewer #2: Yes

4. Is the manuscript presented in an intelligible fashion and written in standard English?

Reviewer #1: Yes

Reviewer #2: Yes

5. Review Comments to the Author

Reviewer #1: Section 1 and 2 are very lengthy. I would suggest to merge these sections under a single heading of Introduction and limit it to two to three pages. Latest references should be incorporated. . Plastic production data of 2015 is mentioned. It is suggested to provide data of 2019.

Reviewer #2: 1. There are two “The” in the title of the manuscript as highlighted above, which seems to be inappropriate. Authors are advised to keep only one.

2. Initial lines of the abstract are well written but there are no statements of findings. The results of work should be written in the abstract to make it more prominent.

3. All the keywords are in upper case. The authors are advised to write them in lower case.

4. The list of abbreviations or symbols used should be placed either at beginning or at the end of manuscript.

5. Legend of figure-1 should contain “that”. It should be like “Factors influencing sustainable purchase decisions”

6. The sections like “practical implications” and Limitations and future prospective” presented in page-18 are not well organized. The authors are advised to reorganize and present the sections before the conclusion.

7. Most of the references are found to be very old. Authors should refer and cite some latest papers related to their research.

Minor revisions given above must be incorporated in the manuscript for consideration.

6. PLOS authors have the option to publish the peer review history of their article (what does this mean?). If published, this will include your full peer review and any attached files.

Reviewer #1: Yes: Dr. Latika Bhatia, Atal Bihari Vajpayee University, Bilaspur, India

Reviewer #2: No

---

## [Author Response · Author response to Decision Letter 0]

28 Aug 2020

COMMENTS TO REVIEWERS

Reviewer 1.

1. Section 1 and 2 are very lengthy. I would suggest to merge these sections under a single heading of Introduction and limit it to two to three pages.

In order that the text not to be so extensive, the introductory chapter was modified, removing content, reducing it to a page and a half. The second and third sections have been merged, also cutting their length.

2. Latest references should be incorporated.

The references of the work have been updated, including the following:

• Blomsma, F. and G. Brennan (2017). The emergence of circular economy: a new framing around prolonging resource productivity. Journal of Industrial Ecology, 21 (3), pp. 603-614.

• Hadladakis J.N. and F. Iacovidou (2018). Closing the loop on plastic packaging materials: What is quality and how does it affect their circularity? Science of the Total Environment 630 (2018) 1394–1400.

• Mugge, R. (2018), Product Design and Consumer Behavior in a Circular Economy, Sustainabilty, 2018, 10 (10, pp. 1-4.

• Nuñez-Cacho, P. Górecki, J. Molina-Moreno, V and F.A. Corpas-Iglesias (2018), What Gets Measured, Gets Done: Development of a Circular Economy Measurement Scale for Building Industry. Sustainability 2018, 10, 2340.

• Prieto-Sandoval, C. Jaca and M. Ormazabal (2018), Towards a consensus on the circular economy. Journal of Cleaner Production, 179. 605-615.

3. Plastic production data of 2015 is mentioned. It is suggested to provide data of 2019.

The data on world plastic production for 2015 has been updated to 2018.

REVIEWER 2

1. There are two “The” in the title of the manuscript as highlighted above, which seems to be inappropriate. Authors are advised to keep only one.

The title has been revised with a translator, including now just one “the”. So, the final title is “Plastics and Sustainable Purchase Decisions in a Circular Economy: The Case of Dutch Food Industry”

2. Initial lines of the abstract are well written but there are no statements of findings. The results of work should be written in the abstract to make it more prominent.

Several new lines have been added to the abstract explaining the results obtained.

3. All the keywords are in upper case. The authors are advised to write them in lower case.

The keywords has been modified according the suggestion of the reviewer.

4. The list of abbreviations or symbols used should be placed either at beginning or at the end of manuscript.

The list of abbreviations has been added at the end of the paper.

5. Legend of figure-1 should contain “that”. It should be like “Factors influencing sustainable purchase decisions”

Legend of figure 1 has been modified to “Figure 1: Factors influencing sustainable purchase decisions”

6. The sections like “practical implications” and Limitations and future prospective” presented in page-18 are not well organized. The authors are advised to reorganize and present the sections before the conclusion.

The section on practical implications and limitations has been reorganized, placing them before the conclusion.

7. Most of the references are found to be very old. Authors should refer and cite some latest papers related to their research. Minor revisions given above must be incorporated in the manuscript for consideration.

We´ve added new references of the last years:

• Blomsma, F. and G. Brennan (2017). The emergence of circular economy: a new framing around prolonging resource productivity. Journal of Industrial Ecology, 21 (3), pp. 603-614.

• Hadladakis J.N. and F. Iacovidou (2018). Closing the loop on plastic packaging materials: What is quality and how does it affect their circularity? Science of the Total Environment 630 (2018) 1394–1400.

• Mugge, R. (2018), Product Design and Consumer Behavior in a Circular Economy, Sustainabilty, 2018, 10 (10, pp. 1-4.

• Nuñez-Cacho, P. Górecki, J. Molina-Moreno, V and F.A. Corpas-Iglesias (2018), What Gets Measured, Gets Done: Development of a Circular Economy Measurement Scale for Building Industry. Sustainability 2018, 10, 2340.

• Prieto-Sandoval, C. Jaca and M. Ormazabal (2018), Towards a consensus on the circular economy. Journal of Cleaner Production, 179. 605-615.

---

## [Editor Report · Decision Letter 1]

16 Sep 2020

Plastics and Sustainable Purchase Decisions in a Circular Economy: The Case of Dutch Food Industry

PONE-D-20-10526R1

Dear Dr. Pedro,

We’re pleased to inform you that your manuscript has been judged scientifically suitable for publication and will be formally accepted for publication once it meets all outstanding technical requirements.

Kind regards,

Prakash Kumar Sarangi, PhD

Academic Editor

PLOS ONE

Additional Editor Comments (optional):

Accepted
---

## [Editor Report · Acceptance letter]

18 Sep 2020

PONE-D-20-10526R1

Plastics and Sustainable Purchase Decisions in a Circular Economy: The Case of Dutch Food Industry

Dear Dr. Núñez-Cacho:

I'm pleased to inform you that your manuscript has been deemed suitable for publication in PLOS ONE. Congratulations! Your manuscript is now with our production department.

Kind regards,

on behalf of

Dr. Prakash Kumar Sarangi 

Academic Editor

PLOS ONE